# A short version of the Alcohol Consumption Consequences Evaluation Scale (ACCE10)

**María-Dolores Sancerni-Beitia[1], Patricia Motos-Sellés[2], José-Antonio Giménez-Costa[2], María-Teresa Cortés-Tomás[2]***

1 Department of Methodology of the Behavioral Sciences, Faculty of Psychology, University of Valencia. Valencia, Spain, 2 Department of Basic Psychology, Faculty of Psychology, University of Valencia, Valencia, Spain

* Maria.T.Cortes@uv.es

**Data Availability Statement:** https://zenodo.org/doi/10.5281/zenodo.12542942.

**Funding:** The author(s) received no specific funding for this work.

## Abstract

A brief version of the Alcohol Consumption Consequences Evaluation Scale (ACCE) [38] was developed to promptly detect possible risks related to alcohol consumption, such as Binge Drinking (BD), in university students. Using the "snowball" method, a sample of 595 students aged 18 to 20 (65.4% women) from the University of Valencia (Spain) was obtained during the 2019–2020 academic year. Items with the highest values of the discrimination parameter in the original version (ACCE) were selected and the Rasch model was applied. To verify the usefulness of this version, ROC analyses were conducted separately for men and women using the Audit score as the criterion. In the overall sample, the analysis had an area of 0.812 (SE = 0.018). In men, the area was 0.796 (SE = 0.032) and for women, it was 0.823 (SE = 0.021). In addition, a logistic regression analysis was performed, using a cut-off point of 3 based on the ROC analysis, to assess the utility of this version in classifying BD and non-BD. The odds ratio was 3.812 (p = 0.000), correctly classifying 89.2% of the young people and indicating that the probability of engaging in BD is 3.8 times higher for individuals obtaining more than 3 points on this scale. This result confirms the usefulness of this brief version (ACCE10) as a screening tool for early intervention, especially in clinical or university settings, since it allows young people to be situated within a range of severity according to their consumption patterns. Furthermore, it may help stop the progression of the addictive process, create awareness of the need for change, and facilitate access to the most suitable interventions.

## Introduction

Over the last two decades, the scientific literature and international health organizations have warned that the intake of large amounts of alcohol is becoming a common and standardized practice among university students [1–6]. In both Spain and other European countries, the prevalence of risky consumption has increased [7], or remains at high levels. Busto-Miramontes et al. [8] concluded that approximately 55% of all university students of both sexes consistently drank alcohol at high-risk levels during the three evaluated periods between 2005

**Competing interests:** The authors have declared that no competing interests exist.

and 2016. On an international scale [9], similar results have been found in Italy, where it was found that 53.3% of the students are high-risk drinkers according to the AUDIT test. In the US, 38% of all university students aged 18 to 22 recognized having engaged in the risky alcohol consumption practice referred to as Binge Drinking (BD) over the previous month, as compared to 33% of their non-student peers [10].

These risky alcohol consumption patterns have numerous biopsychosocial consequences [11–14]. They also create a significant burden for society, not only in terms of health but also on a social and economic level [5, 15, 16]. All of these costs justify the need to identify which university students are at higher risk for the harmful consequences related to their alcohol intake.

Ham and Hope [17] expressed concern about the lack of consensus in defining risk consumption in this population. Until now, no amount of ingested alcohol or specific frequency of consumption behavior has been used in a consensual manner to identify risk consumption. Therefore, many professionals [18–20] have resorted to directly evaluating negative consequences, without considering the quantity and frequency of consumption.

Different self-report instruments have been used to assess the multiple consequences associated with alcohol consumption [21–35]. Over time, they have all been found to have certain limitations, such as focusing almost exclusively on the analysis of social and personal problems [22, 23], including consequences from the most severe end of the alcohol-associated sequelae spectrum [24], or specifically addressing minors and neglecting certain problems that are typical of university students, such as engaging in drinking and driving or risky sexual relations [25, 26], among others.

The Young Adult Alcohol Consequences Questionnaire (YAACQ) [27] attempted to overcome these aspects. It has been the most frequently used instrument to evaluate a wide range of consequences in this population [27–29]. Its brief version includes 24 items (B-YAACQ) [30] to summarize the different types of problems experienced by young people when they consume alcohol. It has been one of the most commonly administered questionnaires given the ease that it offers health professionals and researchers [31–35]. However, the YAACQ and its different versions do not make it possible to sensitively identify young people engaging in less serious risky consumption. This makes it difficult to detect this risky behavior across the continuum of the addictive process [34]. Furthermore, the usefulness of certain items has been questioned in some cases [32, 35], since they function differently in men and women and may underestimate the severity of the consumption behavior. In the Spanish context, Bravo et al. [36] have revealed an inconsistent association between some YAACQ subscales and the quantity and frequency of consumption variables. The evidence of external validity of the YAACQ scores obtained by these authors [36] is not entirely satisfactory.

The Alcohol Consumption Consequences Evaluation Scale (ACCE) [37] was developed in 2020 to overcome these deficiencies. This 43-item scale covers a wide range of consequences experienced by university students when they engage in risky drinking. It is also a practical tool that considers all young alcohol consumers, regardless of their gender, and includes a continuum of representative consequences in young people engaging in different intensities of risky drinking.

Although in psychometric terms, the ACCE is considered a robust measure, shorter instruments are useful for certain clinical settings where greater agility and speed are necessary to detect high-risk consumption. This may be the case in Primary Care, emergencies, or on-campus healthcare. Shorter scales may greatly facilitate the work of health professionals [30]. The use of brief evaluation scale versions in research is also highly recommended [32], especially when several questionnaires must be administered in combination. And it is of particular value for samples of young university students, since this population is the most likely to

complete long batteries of evaluation instruments given their proximity to university research teams. The reliability of the instruments assessing the quantity of alcohol consumed by university students is questionable due to the clear influence of social desirability. Instrument length may have an impact on the obtained results due to the fatigue and saturation that participants may experience.

Therefore, reducing the number of ACCE items will make it easier to include a brief evaluation measure of the consequences both in future research studies where a broad set of variables are evaluated and in treatment/intervention programs where administration times are limited.

Thus, this study aimed to provide a brief version of the ACCE in order to accurately determine if university students experiencing certain consequences are likely to engage in or develop risky alcohol consumption behavior.

## Materials and method

### Sample

The original version of the ACCE [37] was created with a sample of 601 young people aged 18 to 20 from the University of Valencia. It was evaluated over the first quarter of the 2018–2019 academic year. The mean age was 19.25 (0.79), and 64.1% of the interviewees were women (sample A). This was the sample used in this study to obtain evidence of validity. For the objective of this study, the development of the reduced scale, we used a different sample, which was collected during the first quarter of the 2019–2020 academic year. Here, we evaluated 595 young people (the same ages) attending the university and having a mean age of 18.94 (0.80). Of this sample (B), 65.4% were women.

In both samples, the following individuals were excluded from the study: those declaring to have a diagnosed addiction or psychopathology (depression or anxiety symptoms) or those claiming to be taking medications that could cause or mask the consequences and thereby interfere with the study measures.

In accordance with study requirements, the homogeneity between the two samples was verified to ensure that comparisons could be made.

### Procedure

The same data collection procedure was followed for the two samples. This procedure was divided into two phases. In the first phase, the researchers visited the classrooms of first- and second-year Psychology students, requesting the voluntary collaboration of the students. Those who agreed to participate were called on the following day so that they could complete the questionnaire. These participants were then given brief instructions to recruit peers of the same age from both their own degree program and other programs at the University of Valencia. They were to pass along the contact details of the researchers to those interested in participating. Therefore, in the second phase participants were recruited through the "snowball sampling" technique.

In addition, eight final-year students who were collaborating with the research group were trained in instrument administration, to ensure proper completion of the same. All of them had two guided practice sessions under the tutelage of the authors of this study and the instrument was completed in the presence of one of these eight interviewers.

Before conducting the tests, all participants signed a written consent form, where the objectives of the research were clearly reflected, and the anonymity of the offered data was guaranteed. Sample A participants were not allowed to participate in sample B.

The study was conducted in compliance with Spanish legislation (Organic Law 3/2018, of 5 December) and the code of ethics for research involving human subjects, as outlined by the

University of Valencia Human Research Ethics Committee. The survey used in this study is completely anonymous and respondents cannot be identified. Furthermore, the survey contains an introduction that specifies its objectives and potential benefits offered. It also makes explicit reference to compliance with the current Spanish Data Protection Law. The final part of the introduction includes a paragraph where the individual should indicate their agreement to voluntarily participate in the study.

## Variables and instruments

**Consumption pattern.** The alcohol consumption pattern was recorded through self-reporting in the form of a calendar dating back over the past 6 months. This time interval allowed participants to account for their intermittent consumption, including periods of non-consumption that may even exceed 30 days [38, 39].

The self-report measure is an adaptation of the Timeline Followback (TLFB) by Sobell and Sobell [40] and is used as a clinical and research method to obtain quantitative estimates of consumption. The calendar allows individuals to record, on a daily basis, both the frequency and quantity of their alcohol consumption as well as the time when said consumption took place. In other words, they indicate the start and end times of each intake episode, as well as the number of SBUs (Standard Beverage Unit) consumed. For the SBUs, participants were provided with a figure containing the equivalences between alcoholic beverages, their volume, and the number of respective SBUs. Key dates and events were included in the calendar (holidays and exam periods) to serve as milestones to help facilitate recall.

Based on the information provided by the participants in the TLFB, the following variables were generated:

Maximum SBUs consumed: Of all the consumption episodes, the one having the highest amount of SBUs ingested was selected.

Engaging or not engaging in BD. Participants were classified as BD or non-BD based on the Standard Drinking Units (SDUs) consumed during the maximum consumption episode and the number of hours in which the consumption took place. BD has been measured with one of the most frequently used proposals in research, that of the National Institute on Alcohol Abuse and Alcoholism [NIAAA] [41]. In this case, the grams of alcohol proposed by the original definition were adjusted to the Spanish SBU (1 SBU = 10 gr). Therefore, women who consumed six or more SBUs within a 2-hour interval and men consuming seven or more SBUs were classified as BD.

Frequency of BD: The days of BD over the last 6 months were added together. Given that it is intermittent consumption (that does not take place every day) that is variable over time (for example, it may disappear during exam periods, or increase over holidays) the literature recommends keeping a record for at least the past six months [42–44].

**Alcohol Consumption Consequences Evaluation (ACCE) [9].** This one-dimensional questionnaire contains 43 dichotomous (Yes/No) items covering a wide range of consequences derived from an individual's alcohol consumption over the past year. The items are ordered based on parameters of severity (identifying which consequences warn of especially relevant problems to be considered regarding prevention/intervention) and discrimination (indicating that small differences in the trait are associated with a large difference in the probability of accepting the item). This instrument classifies individuals into one of three risk groups: low, moderate, and high risk. In sample A, Cronbach's alpha was 0.845 while in sample B it was 0.817. The full original instrument may be found at Sancerni et al [37].

**AUDIT.** This instrument (Spanish version validated by Contel, Gual, and Colom [45]) consists of 10 items that measure consumption risk, dependency symptoms, and consequences

associated with consumption. It uses a Likert response scale with five alternatives ranging from 0 to 4. In general, when the sum of the scores of all the items is eight or higher, the individual is considered to engage in risky or harmful alcohol consumption. However, when differentiating the scores based on sex, three consumer subtypes are identified: low-risk drinker (cut-off points of 0–7 in men and 0–5 in women), at-risk drinker (cut-off points of 8–12 in men and 6–12 in women), and drinker with physical-psychic problems and probable alcohol dependence (cut-off point of 13 for both) [46]. In sample A, Cronbach's alpha was 0.773 and in sample B it was 0.694.

## Analysis

Statistical analyses were carried out using IBM-SPSS Statistics 26, Jamovi 2.0. and MPlus 8.8.

An item response theory (IRT) model was used. These models are mathematical functions that relate the probability of providing a specific response to an item with the trait of the responding subject. These functions are logistic. In the original study [37], a two-parameter logistic model was used, characterized by use of the parameters of discrimination (a) and position (b) also called severity or, in maximum execution tests, difficulty. The latter, the position parameter (b), informs on the quantity of trait required of the item to be accepted. Hence its name since it serves to *position* the item along the trait continuum. The discrimination parameter (a) indicates the rate of change in the probability of accepting the item as the trait level increases.

In this study, a one-parameter logistic model (Rasch model) was used and only the position parameter (b) was considered, such that each individual's response to each item depends on the subject's position on the trait continuum. It is more likely that a person will accept an item that demonstrates severity (high parameter b) if they are positioned in this area of the trait. This suggests that each person's trait does not depend on the number and type of items that are answered [47, 48]. This model was selected for the creation of the screening test given its parsimony, since it uses only one parameter and does not require a large sample size.

Since the objective of this work is to attain a reduced scale, the following steps were followed: first, the items displaying the highest values of the discrimination parameter in the original instrument [37] were selected.

Second, a one-parameter logistic model (or Rasch model) was applied to these selected items. For model application, it was necessary to verify the assumption of one-dimensionality, the basis of this model. To do so, the following procedure was carried out with the 10 previously selected items: Sample B was randomly divided into two sub-samples. In the first sub-sample, an exploratory factorial analysis was carried out and in the second, a confirmatory factorial analysis was performed using the WLSMV (Weighted Least Squares Means and Variance adjusted) estimation method, which is based on the polychoric correlation matrix, given that the item responses are dichotomous. Goodness of fit was evaluated using the CFI and TLI indices: values above 0.95 indicate excellent fit [49], and for RMSEA, values below 0.05 indicate excellent fit [50]. The potential existence of differences based on sex was verified and Alpha and Omega were calculated as measures of precision [51].

The Rasch model also involves the verification of a property: the measure can only be considered valid and generalizable if it does not depend on the specific conditions under which it was observed. To verify this, once again, sample B was separated into two random sub-samples. In each of these, the position parameters were estimated, and a simple linear regression analysis was performed.

Third, and having verified the previous, the Rasch model was applied to sample B. The fit of the items to the model was assessed through the Infit (WMS) and Outfit (UMS) statistics,

using values in the interval between 0.8 and 1.2 as the fit criterion [52]. Subsequently, it was determined if differences existed in the total score with respect to sex, and reliability was calculated with Cronbach's alpha and McDonald's omega [51].

Fourth, to obtain evidence of validity, the Audit scale and a specific consumption risk score such as that of BD were used as criteria. To select the cutoff points, an analysis of ROC curves was carried out on sample A using the Audit scores as criteria, calculating sensitivity, specificity and the Youden index. Finally, the score was related to BD on the new scale through logistic regression.

## Results

Table 1 shows the characteristics of the two samples used in this study. Sample A was used to develop the ACCE and sample B was used to perform item reduction. No significant differences were found with regard to sex, in either the Audit or the ACCE scores, indicating that the two samples may be considered homogeneous.

The original analysis was used to select the best items [37], focusing on the interval in which maximum information was obtained. The information function showed the maximum at the theta value = 0.20. Given the two-parameter (discrimination and position) logistic model that was applied, an interval of plus-minus 0.5 (-0.3; 0.7) was established around this value. Items displaying the greatest discrimination parameter were chosen. Once the 10 items were selected with this criterion, the Rasch model assumptions applied to sample B were verified. First, and using a random selection of approximately 50% of the cases from sample B1 (n = 288), the verification of one-dimensionality was performed through an exploratory factorial analysis of principal axes from the matrix of tetrachoric correlations, given that the items are dichotomized. The results revealed data adequacy (KMO = 0.895 and Bartlett<0.000), with a single factor being obtained that explained 39.29% of the variance. Second, a confirmatory factorial analysis was performed on the other half of sample B2 (n = 307), obtaining adequate values of the fit indicators: CFI = 0.965; TLI = 0.956; RMSEA = 0.046 (CI = 0.032–0.059). Reliability was calculated with Cronbach's alpha (0.817) and the Omega coefficient (0.81).

The Rasch model was then applied to these 10 items. This global model revealed adequate fit (MADaQ = 0.052; p = 0.089; Pearson Reliability = 0.699). The results for each item are shown in Table 2.

The items were found to have reliable values with low standard errors. Item-test correlations are high and similar in magnitude, indicating a good level of discrimination.

The curve of the test's information function reveals the measurement accuracy along with the trait. It has its maximum at 2.09 (Fig 1), equivalent to a score of 5 points, which correspond to the middle area of the trait continuum. This indicates that the scale is better at estimating the trait of young people who are close to the maximum.

Specific objectivity is one of the most important properties of the Rasch model. A measure may only be considered valid and generalizable if it does not depend on the specific conditions in which it was obtained. Therefore, the difference between two items does not have to depend

**Table 1. Sample characteristics.**

|  | Sample A | Sample B | Contrast | Significance |
|---|---|---|---|---|
|  | (n = 601) | (n = 595) |  |  |
| Sex | 64.1% women | 65.4% women | $\chi^2 = 0.19$ | p = 0.66 |
| Audit | 6.27 (4.28) | 6.71 (3.92) | t = 1.84 | p = 0.06 |
| ACCE | 20.22 (10.65) | 21.03 (9.96) | t = 1.36 | p = 0.17 |

**Table 2. Rasch analysis for ACCE10.**

| Items | b | S.E. | Infit | Outfit | Rit |
|---|---|---|---|---|---|
| While drinking, I have said or done embarrassing things. | -0.46 | 0.102 | 0.911 | 0.853 | 0.580 |
| I have had some bad times | -1.46 | 0.112 | 1.014 | 1.038 | 0.455 |
| I have felt guilty or ashamed of my drinking | -0.54 | 0.102 | 0.958 | 0.922 | 0.537 |
| When drinking, I have done impulsive things that I regretted later | -0.001 | 0.100 | 0.860 | 0.805 | 0.617 |
| I have felt bad about myself because of my drinking | 0.54 | 0.101 | 1.093 | 1.116 | 0.435 |
| I have found that I needed larger amounts of alcohol to feel any effect, or that I could no longer get drunk on the same amount as a few years ago | 0.02 | 0.100 | 1.130 | 1.173 | 0.417 |
| My drinking has caused someone else to feel embarrassed or ashamed | 0.05 | 0.100 | 0.947 | 0.921 | 0.556 |
| When I drink too much I am unable to remember what happened for long periods of time | 0.26 | 0.101 | 1.034 | 1.049 | 0.483 |
| I have had fights or discussions with friends/family about my drinking | 1.41 | 0.109 | 1.015 | 0.968 | 0.437 |
| I have been very rude, obnoxious or insulting after drinking | 1.32 | 0.108 | 1.016 | 0.953 | 0.450 |

b: location parameter item. S.E.: standard error of measurement. Infit: inlier-sensitive or information-weighted fit. Outfit: outlier-sensitive fit. Rit: item-total correlation.

on the individuals calibrating them and, similarly, the difference between two people on a trait does not necessarily depend on the specific items with which said trait has been estimated.

To empirically test this property, two random subsamples were once again generated (B1 = 280 and B2 = 315), the position parameters of the items in each subsample were estimated, and a simple linear regression was performed between the obtained values. The correlation between both sets of parameters, the ordinate at the origin and the slope of the line, were 0.994, 0.012 and 1.053, respectively. The expected values in a perfect fit would be 1, 0 and 1, respectively. Thus, it may be concluded that the invariance property of the parameters is fulfilled.

The greatest advantage of the Rasch model as compared to the classical theory of tests is the property of joint measurement. In other words, the parameters of the items and of the

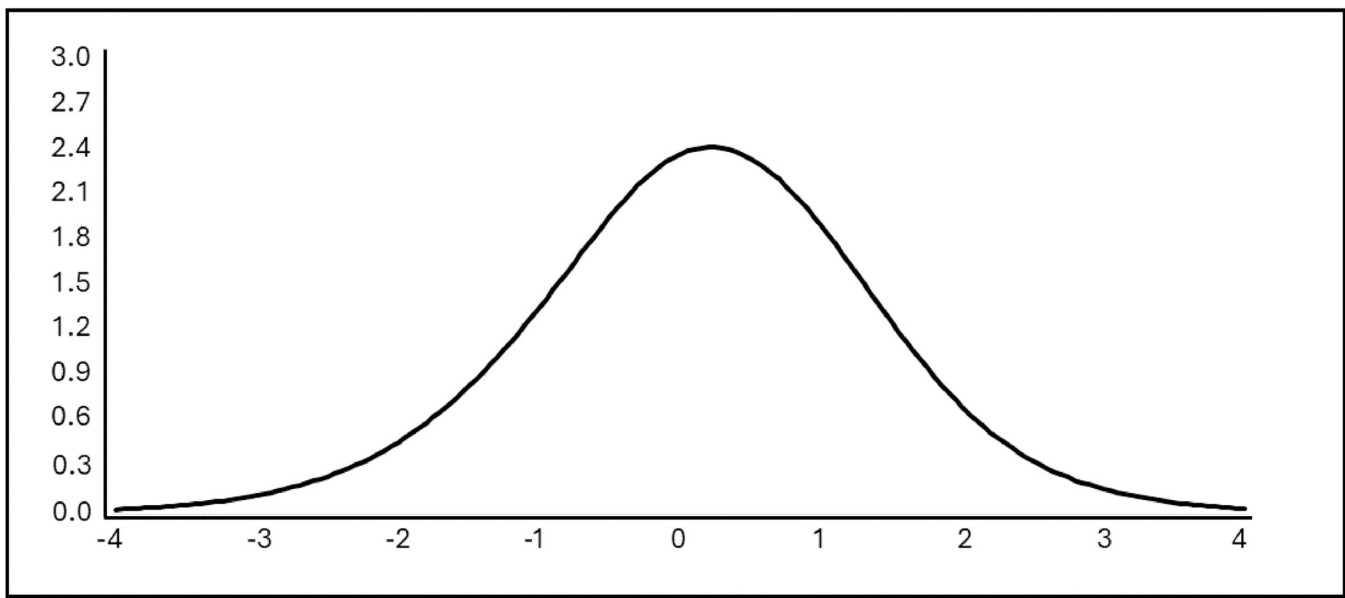

**Fig 1. Information function of the test.**

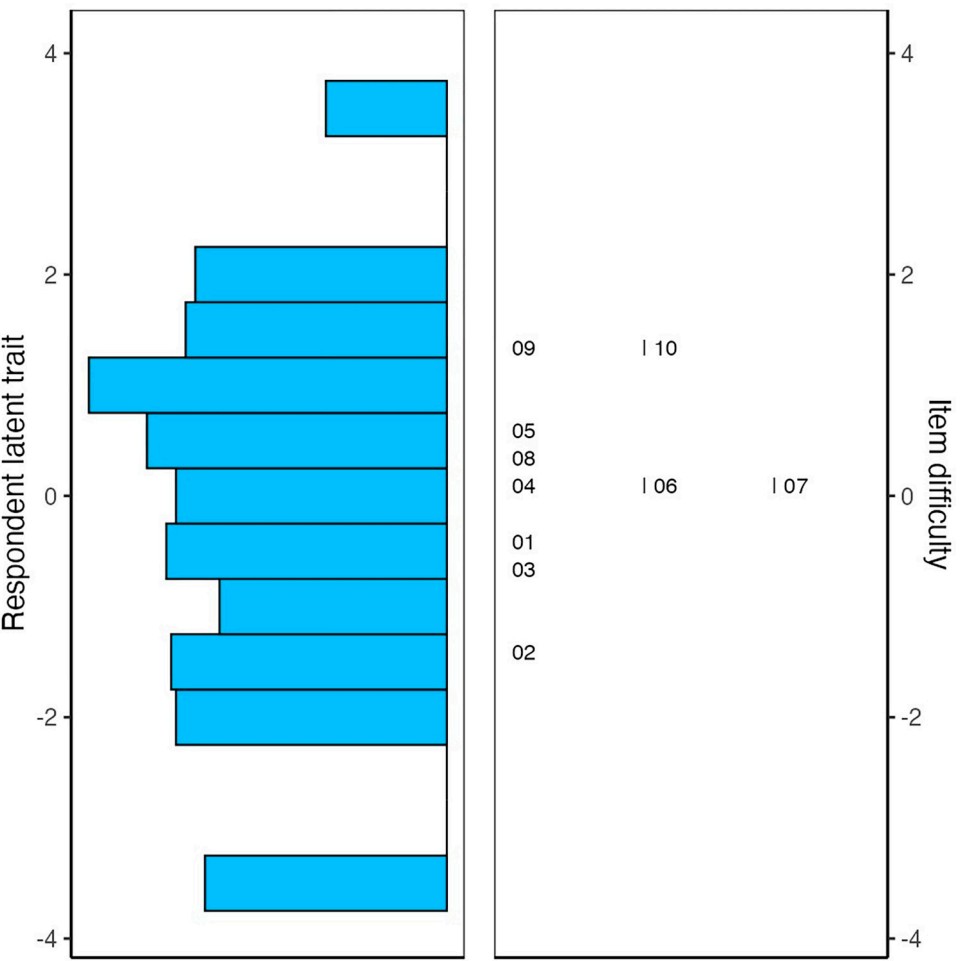

**Fig 2. Wright map: Distribution of items and individuals along the continuum.**

individuals are on the same scale. The joint scaling of items and people can be seen in the map of Fig 2, in which the overlapping of the trait distribution and the parameters is expected to adequately describe the sample.

Once the items from the brief version were calibrated, distinct analyses were carried out on sample B to examine the psychometric properties. No significant differences were found according to sex (t = 0.14; p = 0.88). The effect size was d = 0.01.

Table 3 shows the estimated latent-trait scores for the ACCE10 by each score for the 10 items. The latent trait scores increase as the total score on the ACCE10 increases.

To obtain validity evidence that this version can be useful in classifying alcohol consumers with different risk levels, a ROC analysis was performed using the Audit score as the criterion. The analysis was carried out separately for men and women since the Audit score indicating moderate risk is 5 for women and 7 for men. Table 4 shows the results. For the complete sample, the ROC analysis shows an area of 0.812 (SE = 0.018) with a 95% confidence interval between 0.776 and 0.846. In the case of men, the area was 0.796 (SE = 0.032) with a 95% confidence interval between 0.733 and 0.959. Finally, for women, the area was 0.823 (SE = 0.021) with a 95% confidence interval between 0.781 and 0.865.

The cut-off point was established at 3 for the ACCE10 and a logistic regression was conducted to assess its usefulness in classifying groups of BD/non-BD. The odds ratio was 3.812

**Table 3. Correspondence of the ACCE10 observed scores to latent trait scores.**

| ACCE10 score | Theta | CSEM |
|---|---|---|
| 0 | -3.00 | 1.30 |
| 1 | -2.00 | 0.93 |
| 2 | -1.50 | 0.83 |
| 3 | -1.00 | 0.74 |
| 4 | -0.50 | 0.70 |
| 5 | 0.00 | 0.69 |
| 6 | 0.50 | 0.70 |
| 7 | 1.00 | 0.74 |
| 8 | 2.00 | 0.92 |
| 9 | 3.00 | 1.30 |
| 10 | 3.50 | 1.63 |

Theta = estimated latent-trait score for this raw score on the ACCE10; CSEM = Conditional standard error of measurement.

(p = 0.000), correctly classifying 89.2% of the young people. This suggests that the probability of binge drinking is 3.8 times higher for individuals scoring more than 3 points on the ACCE10. This makes it a very useful screening instrument for early intervention.

## Discussion/Conclusions

The objective of this work was to obtain a brief version of the ACCE (ACCE10), without sacrificing its psychometric suitability, that is capable of identifying the presence of high-risk drinking behavior, through the use of a small number of consequences experienced by young university students when consuming alcohol.

The results allow us to conclude that the 10-item brief version obtained in this study is capable of predict a high percentage of young people who engage in BD (as a type of risk consumption) based on their scores on different consequences derived from their consumption. It should also be noted that, like the long version [37], it uses the same cut-off points for men

**Table 4. Summary of the statistics for the Receiver Operating Characteristic (ROC) analysis to determine cut-off values for the Alcohol Consumption Consequences Evaluation brief form.**

| Cut-off value | Total sample | | | Males | | | Females | | |
|---|---|---|---|---|---|---|---|---|---|
| | Audit cut-off value >5 for females and >7 for males | | | Audit cut-off value >7 | | | Audit cut-off value >5 | | |
| | S | E | Y | S | E | Y | S | E | Y |
| 1 | 0.96 | 0.23 | 0.19 | 0.95 | 0.24 | 0.19 | 0.96 | 0.24 | 0.20 |
| 2 | 0.94 | 0.35 | 0.29 | 0.93 | 0.36 | 0.29 | 0.95 | 0.36 | 0.31 |
| 3 | 0.91 | 0.48 | 0.39 | 0.92 | 0.36 | 0.28 | 0.90 | 0.51 | 0.41 |
| 4 | 0.85 | 0.59 | 0.44 | 0.86 | 0.57 | 0.43 | 0.84 | 0.59 | 0.43 |
| 5 | 0.78 | 0.67 | 0.45 | 0.81 | 0.66 | 0.47 | 0.78 | 0.68 | 0.46 |
| 6 | 0.72 | 0.79 | 0.51 | 0.75 | 0.76 | 0.51 | 0.71 | 0.81 | 0.52 |
| 7 | 0.56 | 0.88 | 0.44 | 0.54 | 0.85 | 0.39 | 0.57 | 0.91 | 0.48 |
| 8 | 0.42 | 0.93 | 0.35 | 0.41 | 0.92 | 0.33 | 0.43 | 0.95 | 0.38 |
| 9 | 0.27 | 0.96 | 0.23 | 0.28 | 0.94 | 0.22 | 0.27 | 0.97 | 0.24 |
| 10 | 0.09 | 0.98 | 0.07 | 0.11 | 0.97 | 0.08 | 0.08 | 0.99 | 0.07 |

S: sensitivity. E: specificity. Y: Youden's index

and women, thereby resolving the limitation of previous scales in representing consequences experienced by both genders in a balanced manner [32, 35].

Regarding professional practice, it can be concluded that the clinical utility of the ACCE10 lies in its ability to help professionals quickly identify young people who could be identified as at risk, based on the consequences that they report experiencing after consuming alcohol. The precise detection of the problems experienced by this group when they consume alcohol through evaluation instruments such as this scale, in both clinical and university settings, could significantly slow down the progression of the addictive process in this population of consumers, generating awareness of the need for change, and facilitating access to the most appropriate interventions in each case [53].

Another advantage of the ACCE10 is that it covers one of the deficiencies pointed out by Falk et al. [18] and Kirouac and Witkiewitz [20] regarding the use of consequences as a sensitive measure over alcohol quantity indicators when carrying out interventions.

Certain limitations of the study should be highlighted, including the fact that the sample was obtained only from University of Valencia students. Furthermore, there is an imbalance in the proportion of male and female participants, although this imbalance is lower than in other studies validating assessment instruments in university students [54]. Therefore, the findings of this study should be replicated using larger groups having a more balanced gender distribution. This would improve the generalization of the utility of this instrument.

It may also be interesting to administer the scale to other young populations who do not attend university and who also admit to engaging in intense alcohol consumption (although to a lesser extent) [10]. In this way, it would be possible to determine if the order of the item severity and the scale properties are maintained in other groups.

It should also be noted that the sample's age range is restricted to first- and second-year students (aged 18 to 20). There is no information available on the functioning of the items and the scale for final-year university students or postgraduate/master's students. Longitudinal studies could be carried out to offer more conclusive results analyzing whether alcohol consumption severity changes across the university trajectory according to development and social context. This type of study would also provide results on the ACCE10 test-retest reliability, the invariance of the item parameters over time, and the sensitivity of individual items to changes in alcohol consumption.

Performing this type of prospective study would make it possible to generalize on the usefulness of the ACCE10 as a research instrument and as a predictive and identifying measure of problems associated with alcohol consumption.

## Author Contributions

**Conceptualization:** María-Dolores Sancerni-Beitia, Patricia Motos-Sellés, José-Antonio Giménez-Costa, María-Teresa Cortés-Tomás.

**Formal analysis:** María-Dolores Sancerni-Beitia, María-Teresa Cortés-Tomás.

**Writing – original draft:** María-Dolores Sancerni-Beitia, Patricia Motos-Sellés, José-Antonio Giménez-Costa, María-Teresa Cortés-Tomás.

**Writing – review & editing:** María-Dolores Sancerni-Beitia, Patricia Motos-Sellés, José-Antonio Giménez-Costa, María-Teresa Cortés-Tomás.

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
