## [Decision Letter · Decision Letter 0]

23 Feb 2024

PONE-D-23-12091A short version of the Alcohol Consumption Consequences Evaluation Scale (ACCE)PLOS ONE

Dear Dr. María-Teresa,

Thank you for submitting your manuscript to PLOS ONE. After careful consideration, we feel that it has merit but does not fully meet PLOS ONE’s publication criteria as it currently stands. Therefore, we invite you to submit a revised version of the manuscript that addresses the points raised during the review process to the extent possible; obviously, it is impossible to go back in time and introduce diaries for tracking the amount alcohol consumed.

We look forward to receiving your revised manuscript.

Kind regards,

Frantisek Sudzina

Academic Editor

PLOS ONE

Journal Requirements:

Reviewers' comments:

Reviewer's Responses to Questions

**Comments to the Author**

1. Is the manuscript technically sound, and do the data support the conclusions?

Reviewer #1: Partly

Reviewer #2: Partly

2. Has the statistical analysis been performed appropriately and rigorously? 

Reviewer #1: Yes

Reviewer #2: Yes

3. Have the authors made all data underlying the findings in their manuscript fully available?

Reviewer #1: No

Reviewer #2: Yes

4. Is the manuscript presented in an intelligible fashion and written in standard English?

Reviewer #1: No

Reviewer #2: Yes

5. Review Comments to the Author

Reviewer #1: The authors set out to develop a short version of an existing longer Alcohol Consumption Consequences Evaluation Scale. It seems that this is a valid research task that could be reported in Plos One, and the data and the analysis seems mostly adequate as well. However, the write-up of the paper is problematic in many ways, and it is very hard for the reader to follow what is being done. I will detail the main issues below.

Main problems

1. The quality or the clarity of the writing is not good enough to allow the reader to follow and understand what is being done and why. Examples are detailed below.

2. The authors develop an instrument consisting of questions that can be asked from young people so that the extent of alcohol-related harm they experience could be understood. However, in the whole report the authors do not inform the readers about the content of the questions included either in the longer original ACCE instrument or the brief version of 10 questions, which is the outcome from this report. I don’t think this is acceptable.

3. The conclusions drawn are to some extent awkward (which raises questions also about the formulation of the aim) or not supported by the analysis or what I think has been done in the analyses.

- “The results allow us to conclude that the 10-item short version obtained in the present study is capable of identifying a high percentage of young people who engage in BD based on their scores in different consequences derived from their consumption.” and the objective being stated as identifying “the presence of a high risk of excessive alcohol consumption in the university population, including BD”. If the measure against which the success is compared is binge drinking, why not just ask about binge drinking? It is not sensible to ask 10 questions to get a proxy for something that could be asked with one question. The idea seems upside-down: the reason why we are interested in binge drinking is that it can cause alcohol-related harm. Therefore, the measurement of alcohol-related harm is of essence in itself, not just as a proxy for risky drinking. This formulation bothered me only in the discussion and not in the introduction, so it may be more of a question of expressions and formulation of sentences rather than that what was done would be flawed.

- Gender differences: it is said that the brief ACCE “uses the same cut-off points for men and women, which resolves the limitation that previous scales had on the difficulty of representing consequences experienced by both genders in a balanced way”. Merely having a same cut-off point for something in no way guarantees it that the measure represents consequences experienced by both genders in a balanced way. This should be evaluated on the basis of the questions asked, which the reader is currently not capable of doing.

4. Specification of aims. Relating to point 3a above, I think there is a need to write more specific aims or research questions in the introduction. Technical methods belong to the methods section, but the general approach of what the authors set out to do and why belong to the introduction. The first task seems to be to find a subset of questions among the original ACCE questions that efficiently capture the essence of the trait. The second seems to be to find out cut-points for the brief instrument, and this is done by comparison against AUDIT (& shortly why the authors think this is a justified approach to take) and, third, the authors examine the performance of the new brief instrument when comparing against a binge drinking measure (& shortly why the authors think this is a justified approach to take). By adding this the methods and analysis and presentation of results would be easier to follow, and justifications for the approach would not need to be explained in methods section. Also, this would allow better explanation of what has perhaps already been done in the original study and what is a new contribution (see a further comment on this below).

5. Measurement: the authors write: “The pattern of alcohol consumption was registered using a self-report in the format of a complete calendar of the last 6 months.” This would be perfect if respondents were actually capable of remembering all their drinking occasions in the past 6 months. If someone drinks 5 times a week, it makes 130 drinking occasions. Even 2 times a week makes 52 occasions. In practice, people who drink regularly have difficulties remembering drinking occasions beyond one week. Only if drinking is an exceptional activity for a person can they remember back 6 months. This problem has not been critically discussed in the paper.

6. Measurement 2 (this is not merely a detail, but it is not something I would expect the authors to change in their manuscript, either): the NIAAA definition of binge drinking was used to decide if the maximum consumption constituted binge drinking. This is somewhat problematic. If someone has drunk 10 drinks in 5 hours, this is not considered binge drinking. In another time they may have drunk 8 drinks (not their maximum) in 2 hours – this would be considered binge drinking if it was included, but only the maximum-drinks occasion was included. Overall, the NIAAA 2-hour time frame is arbitrary. A less arbitrary way to analyse the Spanish data would be using eBAC, estimated blood alcohol concentration at the end of the drinking episode (see for example Hustad & Carey https://doi.org/10.15288/jsa.2005.66.130) if they have recorded the respondents’ weight.

7. Results, Table 4. Why would a cut-off of 3 be chosen and not 6? Is specificity of 0.48 sufficient in the authors’ view?

Issues with write-up

Introduction: The presentation of what has been done previously, the original work on ACCE, the problems with previous work and what is the aim now is not presented logically in a manner that would be easy for the reader to follow: it’s first stated that there are many instrument (without specifying what these are), then some problems in them are mentioned, then the authors present one named instrument (without informing the reader why this is chosen or what status it has in the current study or in the text), with some critique about this measure (some of it is not in an acceptable form ‘external validity of the YAACQ scores obtained by these authors is not entirely satisfactory’ – without explanation or a reference); then authors jump to ACCE (but there is no proper presentation of it: what is this scale? how many items? has someone else used it? It starts “The ACCE” as if the reader is assumed to know it). Without specifying the length of the original ACCE it was hard to easily understand the move from ‘this has been done previously’ to ‘this is the gap in knowledge we are going to address in THIS study’.

Overall, it was at times difficult to follow what was done in the “original study” (ref 9) and what was the new independent contribution here.

- It is difficult to follow whether sample A (the study population in the original study) was also used here with a new sample B. It is stated in the methods section that “For the present study, we used a different sample” (Sample B), but in the results section there are numerous mentions about sample A being used for this and that, and Table 1 lists sample characteristics also for Sample A.

- Are these results using sample A already presented in the original study (ref 9) and just mentioned here, or are these new results? If they are new results, then the methods section needs to be re-written so that the reader understand that there are two samples used in the current paper, one of which was collected for the original study and the other for this study.

- The results section: “The original analysis was used to choose the best items (9), focusing on the interval in which maximum information was obtained. The information function showed the maximum at the value theta=0.20. Given the two-parameter logistic model applied (discrimination and position), an interval of plus-minus 0.5 (-0.3; 0.7) was established around this value, and items that showed the greatest discrimination parameter were chosen. Once the 10 items were selected with this criterion, the Rasch model assumptions applied to sample B were checked (…)” – so is it to be understood that all this has already been reported in the previous study? If the item reduction was already done previously, what is the new contribution, or the difference to the previous study? (It is written “Once the 10 items were selected…” – was it decided already beforehand that 10 will be selected, or where did the 10 come from? If it was decided beforehand, also this should be motivated in the introduction)

Analysis: the paragraph about item selection was not clear. The qualities of the procedure are explained, but it is not explained (at least not at first / not where the reader would expect this to happen, i.e. at first) what the procedure itself is. What does it mean that item response theory was used, for those who don’t know this already? What does a one-parameter logistic model or Rasch model do? First explain what this model is and only then (if needed at all) explain why it is better than alternatives. The one parameter is “position or difficulty” – difficulty of what? “For the selection of items, the criterion of greater discrimination was used in the original study, in which a two parameter logistic model was applied for calibration.” – how does this relate to the current study, is it just a ‘nice to know’ information (in which case it could be deleted).

Thereafter, “an exploratory factorial analysis was conducted with sample B to verify dimensionality”. Please explain to the reader what the aim is in verifying dimensionality or what it means. References are needed after statements like “Alpha and Omega were calculated as measures of precision”. Then, “A Rasch model was applied, checking the global fit” – please explain what a Rasch model does and what it is used for.

“[the test’s information function] has its maximum at 2.09 (Fig 1) equivalent to a score of 5 points, which correspond to the middle area of the trait continuum, indicating that the scale is better for estimating the trait of young people who are closed to the maximum” What score is this? Is it the ACCE score of the original score, or a score of the brief instrument? The latter has not been previously mentioned. Why does the score of 5 points indicate that the scare is better for estimating the trait for those close to the maximum?

If two random subsamples of similar size were taken, why is one n=280 and the other n=315?

Figure 2 needs to be better explained.

“Once the items of the brief version were calibrated, different analyses were carried out on sample A to study the psychometric properties. No significant differences were found according to sex (…) The size of the effect was d=0.01” – properties of what? Differences in what? Effect of what on what?

“The Table 3 shows the estimated latent-trait scores for the ACCE10 by each score of the 10 items” “Table 3. Correspondence of the ACCE10 scores to latent trait scores”. What does this mean – what is presented in the table? What is CSEM?

“The odds ratio was 3.812” – odds of what in what group compared to what group?

Reviewer #2: The manuscript “A short version of the Alcohol Consumption Consequences Evaluation Scale (ACCE)” describes a straightforward endeavor to validate a briefer version of the ACCE. The manuscript has many strengths, in my opinion, including focus on a potentially impactful and practical outcome (e.g., a brief version of the scale would have high research and clinical utility), what appears to be a rigorous approach to validate the instrument, and clear and straightforward writing. However, I have some concerns and critiques with/of the current manuscript. I outline each concern below.

1. The statement in the Introduction “Until now, there is no amount of alcohol ingested, nor any specific frequency of consumption behavior, which can be used as a reference to identify problematic consumption” appears to be controversial and perhaps counterintuitive given the later reference to the construct of “binge drinking” (which does have clearly defined amounts). Some further clarity here would help readers to understand the motivation for the study.

2. Justification for the exclusion criteria (e.g., medication use) used for the study is needed to clarify the evidence presented.

3. Some consideration of statistical power in regard to the sample size for the multitude of analyses presented is needed for readers to better understand the results presented.

4. Many readers will want a clearer description and justification for the choice of analyses performed, especially when some varying analytical techniques partially overlap in their purpose.

5. The weaknesses of the study are noted, how much they limit the degree of evidence presented should, in my opinion, be emphasized more throughout the manuscript. For example, the restricted age range, snowball sampling techniques, and restriction to one university all likely severely limit the generalizability of the results.

6. Related to point 4 above, some information about how the results are affected by the race and ethnicity of the samples would improve the clarity of the results presented.

6. PLOS authors have the option to publish the peer review history of their article (what does this mean?). If published, this will include your full peer review and any attached files.

Reviewer #1: No

Reviewer #2: No

---

## [Author Response · Author response to Decision Letter 0]

2 Jul 2024

Reviewer's Responses to Questions

Comments to the Author

1. Is the manuscript technically sound, and do the data support the conclusions? Reviewer #1: Partly // Reviewer #2: Partly

(R) (Following the reviewers’ recommendations, some sections of the article have been modified to clarify how the screening instrument was extracted. The conclusions derived from these analyses have also been revised.

2. Has the statistical analysis been performed appropriately and rigorously? Reviewer #1: Yes // Reviewer #2: Yes

3. Have the authors made all data underlying the findings in their manuscript fully available? Reviewer #1: No // Reviewer #2: Yes

(R) The journal’s rules regarding this have been followed.

4. Is the manuscript presented in an intelligible fashion and written in standard English? Reviewer #1: No // Reviewer #2: Yes

(R) The revised article has been sent to a company of native translators. A certificate issued by this company has been included.

5. Review Comments to the Author

(R) We would like to thank the reviewers for all their comments. These have greatly improved the quality of the final manuscript. A table with the comments made and the response to them is included below.

Reviewer #1: The authors set out to develop a short version of an existing longer Alcohol Consumption Consequences Evaluation Scale. It seems that this is a valid research task that could be reported in Plos One, and the data and the analysis seems mostly adequate as well. However, the write-up of the paper is problematic in many ways, and it is very hard for the reader to follow what is being done. I will detail the main issues below.

Main problems

1. The quality or the clarity of the writing is not good enough to allow the reader to follow and understand what is being done and why. Examples are detailed below.

(R) The text has been reviewed in its entirety, modifying certain expressions and correcting any errors. Certification from the translation and revision company has been included.

(R) Furthermore, distinct sections of the article have been reviewed in an attempt to clarify what has been done and why.

2. The authors develop an instrument consisting of questions that can be asked from young people so that the extent of alcohol-related harm they experience could be understood. However, in the whole report the authors do not inform the readers about the content of the questions included either in the longer original ACCE instrument or the brief version of 10 questions, which is the outcome from this report. I don’t think this is acceptable.

(R) In accordance with the accurate comment made by the reviewer, the items from the ACCE10 have been included in Table 2. 

(R) Furthermore, indications have been included detailing how to access the full original instrument (ACCE) in the method section.

3. The conclusions drawn are to some extent awkward (which raises questions also about the formulation of the aim) or not supported by the analysis or what I think has been done in the analyses. - “The results allow us to conclude that the 10-item short version obtained in the present study is capable of identifying a high percentage of young people who engage in BD based on their scores in different consequences derived from their consumption.” and the objective being stated as identifying “the presence of a high risk of excessive alcohol consumption in the university population, including BD”. If the measure against which the success is compared is binge drinking, why not just ask about binge drinking? It is not sensible to ask 10 questions to get a proxy for something that could be asked with one question. The idea seems upside-down: the reason why we are interested in binge drinking is that it can cause alcohol-related harm. Therefore, the measurement of alcohol-related harm is of essence in itself, not just as a proxy for risky drinking. This formulation bothered me only in the discussion and not in the introduction, so it may be more of a question of expressions and formulation of sentences rather than that what was done would be flawed

(R) We apologize for not having been clearer. We have modified some paragraphs from the end of the introduction and the discussion sections in order to express this idea more clearly. 

(R) The need for an instrument to obtain an approximation of something (excessive consumption) that can be done with a single question has been questioned. The objective is not to detect excessive consumption, it is to obtain a scale of consequences permitting its prediction, which may be useful for prevention. 

- Gender differences: it is said that the brief ACCE “uses the same cut-off points for men and women, which resolves the limitation that previous scales had on the difficulty of representing consequences experienced by both genders in a balanced way”. Merely having a same cut-off point for something in no way guarantees it that the measure represents consequences experienced by both genders in a balanced way. This should be evaluated on the basis of the questions asked, which the reader is currently not capable of doing.

(R) The questions forming the ICCE10 have been included in the article’s text in Table 2, providing the reader with all of the necessary information.

4. Specification of aims. Relating to point 3a above, I think there is a need to write more specific aims or research questions in the introduction. Technical methods belong to the methods section, but the general approach of what the authors set out to do and why belong to the introduction. The first task seems to be to find a subset of questions among the original ACCE questions that efficiently capture the essence of the trait. The second seems to be to find out cut-points for the brief instrument, and this is done by comparison against AUDIT (& shortly why the authors think this is a justified approach to take) and, third, the authors examine the performance of the new brief instrument when comparing against a binge drinking measure (& shortly why the authors think this is a justified approach to take). By adding this the methods and analysis and presentation of results would be easier to follow, and justifications for the approach would not need to be explained in methods section. Also, this would allow better explanation of what has perhaps already been done in the original study and what is a new contribution (see a further comment on this below).

(R) No sub-sections are detailed in the objectives since, during the instrument development process, results are presented to support the instrument as a whole. Once the questions have been selected, different types of evidence are obtained, but these are not objectives in themselves. Rather, they are steps in the process of validating the scale scores.

(R) What we have done, in response to both reviewers, is carefully draft all of the steps followed, along with the analysis performed in each of them, in the Analysis section

5. Measurement: the authors write: “The pattern of alcohol consumption was registered using a self-report in the format of a complete calendar of the last 6 months.” This would be perfect if respondents were actually capable of remembering all their drinking occasions in the past 6 months. If someone drinks 5 times a week, it makes 130 drinking occasions. Even 2 times a week makes 52 occasions. In practice, people who drink regularly have difficulties remembering drinking occasions beyond one week. Only if drinking is an exceptional activity for a person can they remember back 6 months. This problem has not been critically discussed in the paper.

(R) Our previous work (Cortés et al., 2016, 2017) justifies the need to adjust the time period in which the pattern of consumption is assessed to the characteristics of the university population. Given that it is an intermittent behaviour (Heather and Stockwell, 2004; Ministry of Health and Consumer Affairs, 2008; Townshend and Duka, 2009; Weissenborn and Duka, 2003), that is sensitive to the particularities of the time period in which students are living (exam period, holidays, etc.), small variations in the time frame selected to assess their consumption behaviour may produce significant changes in consumption rates, leading to potential errors in the detection (or not) of possible risk to consumers. The recommendation from the literature is to use a time frame that describes the variability in their drinking behaviour, with as little recall distortion as possible, like the 6-month time frame (Knudsen and Skogen, 2015; Townshend and Duka, 2009).

(R) In addition, as specified in the text, the TLFB allows us to include a reminder of the dates indicated in the evaluated months. This facilitates memory of the individual completing the self-registration.

6. Measurement 2 (this is not merely a detail, but it is not something I would expect the authors to change in their manuscript, either): the NIAAA definition of binge drinking was used to decide if the maximum consumption constituted binge drinking. This is somewhat problematic. If someone has drunk 10 drinks in 5 hours, this is not considered binge drinking. In another time they may have drunk 8 drinks (not their maximum) in 2 hours – this would be considered binge drinking if it was included, but only the maximum-drinks occasion was included. Overall, the NIAAA 2-hour time frame is arbitrary. A less arbitrary way to analyse the Spanish data would be using eBAC, estimated blood alcohol concentration at the end of the drinking episode (see for example Hustad & Carey https://doi.org/10.15288/jsa.2005.66.130) if they have recorded the respondents’ weight.

(R) The time period used is based on the NIAAA (2004) definition. We agree that this consumption time frame should be reviewed in order to assess it more appropriately.

7. Results, Table 4. Why would a cut-off of 3 be chosen and not 6? Is specificity of 0.48 sufficient in the authors’ view?

(R) We understand that the reviewer is referring to 6 as the cut-off point since the Youden index is the highest at that point. Here, we make the following considerations: The Youden index should not be used in isolation as it is an index that provides equal weight to sensitivity and specificity. Cut-off points should not be based solely on statistical criteria; in many cases, it is more important to base them on what the test is intended to achieve: greater sensitivity or greater specificity. In general, screening tests should have high sensitivity to find all people with the problem, since failing to find them means failing to treat them. Higher specificity is sought in tests that confirm a diagnosis, when there is interest in people who do not have the problem or when misdiagnosing a person who does not really have the problem may have serious consequences. In our case, we are obviously seeking high sensitivity.

Issues with write-up

(R) The article has been reviewed by a translation and editing company.

Introduction: The presentation of what has been done previously, the original work on ACCE, the problems with previous work and what is the aim now is not presented logically in a manner that would be easy for the reader to follow: it’s first stated that there are many instrument (without specifying what these are), then some problems in them are mentioned, then the authors present one named instrument (without informing the reader why this is chosen or what status it has in the current study or in the text), with some critique about this measure (some of it is not in an acceptable form ‘external validity of the YAACQ scores obtained by these authors is not entirely satisfactory’ – without explanation or a reference); then authors jump to ACCE (but there is no proper presentation of it: what is this scale? how many items? has someone else used it? It starts “The ACCE” as if the reader is assumed to know it). Without specifying the length of the original ACCE it was hard to easily understand the move from ‘this has been done previously’ to ‘this is the gap in knowledge we are going to address in THIS study’.

Overall, it was at times difficult to follow what was done in the “original study” (ref 9) and what was the new independent contribution here.

(R) The introduction section has been revised in order to make it more coherent. The reviewer indicated that the name of certain instruments was not included, aspects which we believe to have been addressed since the reader may consult the bibliography for this purpose. An emphasis is only placed on the instrument that, as indicated in the introduction, has attempted to overcome the problems presented by all the other ones.

(R) We wish to thank the reviewer who indicated the lack of citation of an aspect that is of great importance: the external validity of the YAACQ. This has been corrected. Similarly, we have completed the text presenting the description of the ACCE.

(R) The independent contribution of this work is that it provides a reduced measure that can be used as a screening instrument, as highlighted in the last paragraph of the introduction. Furthermore, in the modified analysis section, the development of the objective of this work is clearly described.

- It is difficult to follow whether sample A (the study population in the original study) was also used here with a new sample B. It is stated in the methods section that “For the present study, we used a different sample” (Sample B), but in the results section there are numerous mentions about sample A being used for this and that, and Table 1 lists sample characteristics also for Sample A.

- Are these results using sample A already presented in the original study (ref 9) and just mentioned here, or are these new results? If they are new results, then the methods section needs to be re-written so that the reader understand that there are two samples used in the current paper, one of which was collected for the original study and the other for this study.

(R) When extracting a screening instrument from another more extensive instrument, it is necessary to include certain statistical data from the original sample (sample A used for the original instrument) such as the distribution of the original sample by sex, and measures in the variables that will be used here, so that it may be observed that the original sample (sample A) and the sample used in the screening instrument (sample B), are similar.

(R) When reviewing the methods section, certain parts have been re-written so that they are clearer, including the sample and analysis subsections.

- The results section: “The original analysis was used to choose the best items (9), focusing on the interval in which maximum information was obtained. The information function showed the maximum at the value theta=0.20. Given the two-parameter logistic model applied (discrimination and position), an interval of plus-minus 0.5 (-0.3; 0.7) was established around this value, and items that showed the greatest discrimination parameter were chosen. Once the 10 items were selected with this criterion, the Rasch model assumptions applied to sample B were checked (…)” – so is it to be understood that all this has already been reported in the previous study? If the item reduction was already done previously, what is the new contribution, or the difference to the previous study? (It is written “Once the 10 items were selected…” – was it decided already beforehand that 10 will be selected, or where did the 10 come from? If it was decided beforehand, also this should be motivated in the introduction)

Analysis: the paragraph about item selection was not clear. The qualities of the procedure are explained, but it is not explained (at least not at first / not where the reader would expect this to happen, i.e. at first) what the procedure itself is. What does it mean that item response theory was used, for those who don’t know this already? What does a one-parameter logistic model or Rasch model do? First explain what this model is and only then (if needed at all) explain why it is better than alternatives. The one parameter is “position or difficulty” – difficulty of what? “For the selection of items, the criterion of greater discrimination was used in the original study, in which a two parameter logistic mode

---

## [Editor Report · Decision Letter 1]

2 Aug 2024

A short version of the Alcohol Consumption Consequences Evaluation Scale (ACCE10)

PONE-D-23-12091R1

Dear Dr. María-Teresa,

We’re pleased to inform you that your manuscript has been judged scientifically suitable for publication and will be formally accepted for publication once it meets all outstanding technical requirements.

Kind regards,

Frantisek Sudzina

Academic Editor

PLOS ONE
---

## [Editor Report · Acceptance letter]

9 Aug 2024

PONE-D-23-12091R1 

PLOS ONE

Dear Dr. Cortés-Tomás, 

I'm pleased to inform you that your manuscript has been deemed suitable for publication in PLOS ONE. Congratulations! Your manuscript is now being handed over to our production team.

Kind regards, 

on behalf of

Dr. Frantisek Sudzina 

Academic Editor

PLOS ONE